# Prediction of Nonalcoholic Fatty Liver Disease Using Noninvasive and Non-Imaging Procedures in Japanese Health Checkup Examinees

**DOI:** 10.3390/diagnostics11010132

**Published:** 2021-01-16

**Authors:** Kenichiro Murayama, Michiaki Okada, Kenichi Tanaka, Chika Inadomi, Wataru Yoshioka, Yoshihito Kubotsu, Tomomi Yada, Hiroshi Isoda, Takuya Kuwashiro, Satoshi Oeda, Takumi Akiyama, Noriko Oza, Hideyuki Hyogo, Masafumi Ono, Takumi Kawaguchi, Takuji Torimura, Keizo Anzai, Yuichiro Eguchi, Hirokazu Takahashi

**Affiliations:** 1Division of Metabolism and Endocrinology, Faculty of Medicine, Saga University, Saga 849-8501, Japan; kenichirom1004@yahoo.co.jp (K.M.); f8388@cc.saga-u.ac.jp (M.O.); sj8833@cc.saga-u.ac.jp (K.T.); chlkqiko@gmail.com (C.I.); sailingxyz94@yahoo.co.jp (W.Y.); y.05211027@gmail.com (Y.K.); f8451@cc.saga-u.ac.jp (T.K.); akiyamat@cc.saga-u.ac.jp (T.A.); akeizo0479@gmail.com (K.A.); 2Department of Hepatobiliary and Pancreatology, Saga Medical Center Koseikan, Saga 840-8571, Japan; ohza-n@koseikan.jp; 3Liver Center, Saga University Hospital, Faculty of Medicine, Saga University, Saga 849-8501, Japan; yadat@cc.saga-u.ac.jp (T.Y.); e6140@cc.saga-u.ac.jp (H.I.); ooedasa@cc.saga-u.ac.jp (S.O.); eguchiyu@me.com (Y.E.); 4Department of Gastroenterology and Hepatology, JA Hiroshima General Hospital, Hatsukaichi 738-8503, Japan; hidehyogo@ae.auone-net.jp; 5Tokyo Women’s Medical University Medical Center East, Internal Medicine, Tokyo 116-8567, Japan; ono.masafumi@twmu.ac.jp; 6Division of Gastroenterology, Department of Medicine, Kurume University School of Medicine, Kurume 830-0011, Japan; takumi@med.kurume-u.ac.jp (T.K.); tori@med.kurume-u.ac.jp (T.T.); 7Department of Clinical Gastroenterology, Eguchi Hospital, Ogi 845-0032, Japan

**Keywords:** nonalcoholic fatty liver disease, ultrasonography, hepatic steatosis index, fatty liver index, fibrosis-4 index, ROC, health checkup

## Abstract

Access to imaging is limited for diagnosing nonalcoholic fatty liver disease (NAFLD) in general populations. This study evaluated the diagnostic performance of noninvasive and nonimaging indexes to predict NAFLD in the general Japanese population. Health checkup examinees without hepatitis virus infection or habitual alcohol drinking were included. Fatty liver was diagnosed by ultrasonography. The hepatic steatosis index (HSI), Zhejiang University (ZJU) index, and fatty liver index (FLI) were determined, and risk of advanced liver fibrosis was evaluated by the fibrosis-4 index. NAFLD was diagnosed in 1935 (28.0%) of the 6927 subjects. The area under the receiver operating characteristic (AUROC) curve of the HSI, ZJU index, and FLI was 0.874, 0.886, and 0.884, respectively. The AUROC of the ZJU index (*p* < 0.001) and FLI (*p* = 0.002) was significantly greater than that for the HSI. In subjects with a high risk of advanced fibrosis, the sensitivity of the HSI, ZJU index, and FLI were 88.8%, 94.4%, and 83.3% with a low cut-off value and the specificity was 98.5%, 100%, and 100% with a high cut-off value. In conclusion, all indexes were useful to diagnose NAFLD in the general Japanese population and in subjects with potentially advanced liver fibrosis.

## 1. Introduction

The increase in metabolic syndrome due to obesity has become a global problem, and Japan is no exception [1,2,3]. Nonalcoholic fatty liver disease (NAFLD) covers a spectrum of liver diseases that range from benign simple steatosis to the hepatic inflammation and fibrosis of nonalcoholic steatohepatitis, cirrhosis, and hepatocellular carcinoma. NAFLD is a hepatic manifestation of metabolic syndrome [4,5]. Therefore, the prognosis of NAFLD is affected not only by liver-related diseases such as cirrhosis and hepatocarcinogenesis, but also by all diseases and conditions that have a common background with NAFLD. These diseases and conditions include obesity, insulin resistance, diabetes, and dyslipidemia [6].

According to previous reports, the survival rate of NAFLD was significantly lower than that of the general population. The most common causes of death in NAFLD in the United States are cardiovascular diseases (CVDs) (25%), extrahepatic malignancies (28%), and liver diseases (13%). Factors related to mortality are age, impaired glucose tolerance, and cirrhosis [7,8,9]. Because NAFLD is a liver disease against the background of lifestyle-related diseases, the prognosis for NAFLD is associated not only with the liver effects, but also with the effects of the progression of other lifestyle-related diseases. These diseases include CVD and chronic kidney disease (CKD) that result from visceral obesity, arteriosclerosis, and diabetes. Multiple large epidemiologic studies have shown that NAFLD is an independent risk factor for the development of CVD [10,11,12]. In addition, CVD mortality in NAFLD was higher than in the general population [13]. NAFLD has also been reported to be a risk factor for CKD independent of metabolic syndrome [14,15,16]. Therefore, diagnosis of NAFLD is important. The prevalence of NAFLD in Japan is reported to be 29.7% and it is estimated that 37.4 million people have NAFLD [17].

Imaging examinations including abdominal ultrasonography are generally used for diagnosing NAFLD; however, it is difficult from the viewpoint of medical economy to test the whole population. In addition, with the influence of COVID-19 [18,19], which has recently spread throughout the entire world, abdominal ultrasound can be risky due to the concentrated contact. It has also been reported that NAFLD is a risk factor for severe COVID-19 infection [20]. Therefore, it is currently required to predict the diagnosis of NAFLD by a noncontact procedure rather than ultrasound examination. By previous reports, there are three indexes for predicting NAFLD: the fatty liver index (FLI), Zhejiang University (ZJU) index, and hepatic steatosis index (HSI) [21,22,23]. However, the results of a direct comparison of the diagnostic performance of these indexes remains unclear. Moreover, it is important to confirm whether the diagnostic performance of these indexes is reliable in patients with mortality risks such as liver fibrosis and diabetes [24,25]. The aim of this study was to examine the diagnostic performance of prediction formulas to identify NAFLD in the general population who underwent health checkups in Japan.

## 2. Materials and Methods

### 2.1. Subjects

This cross-sectional study was conducted with data from 15,785 subjects who received general health checkups in 2009 and 2010 in three Japanese health centers: Eguchi Hospital Health Center in Saga, Kawamura Clinic Health Center in Hiroshima, and Kochi Medical School Hospital in Kochi. This cohort was previously analyzed to identify the prevalence of NAFLD in Japan [17], to identify the reference range for alanine aminotransferase (ALT) level [26], and to investigate the relationship between alcohol intake and NAFLD [27]. The health examination included physical and physiological examinations, abdominal ultrasonography, and blood screening tests. We excluded 5074 subjects with incomplete data and 329 subjects positive for hepatitis B surface antigen or hepatitis C antibody. We also excluded 3455 subjects who were habitual drinkers (male > 30 g/day, female > 20 g/day), who were considered to have alcoholic liver damage. Finally, 6927 subjects were enrolled in this study. All subjects provided written informed consent for the anonymous use of their data in this epidemiological study. The study design was approved by each institutional review board (Saga University, “4 June 2011”; Eguchi Hospital, R1-1 (2019); Hiroshima University, “Eki-241” as Kawamura Clinic Health center; and Kochi University, “23–74”). This study was conducted in accordance with the Declaration of Helsinki.

### 2.2. Physical Examination and Laboratory Tests

Body weight and height were measured, and body mass index (BMI) was calculated as weight (kg) divided by height squared (m^2^). Waist circumference (WC) was measured at the umbilical level. According to criteria established by the Japan Society for the Study of Obesity, visceral adiposity was defined as WC > 85 cm in males and >90 cm in females [28]. Venous blood samples were taken from all subjects following a 12 h overnight fast, and aspartate aminotransferase (AST), ALT, γ-glutamyl transpeptidase (GGT), total cholesterol, high-density lipoprotein cholesterol (HDL-C), low-density lipoprotein cholesterol (LDL-C), triglyceride (TG), hemoglobin A1c (HbA1c), and fasting plasma glucose (FPG) concentrations were measured using standard techniques in the subjects who received a health examination. The diagnosis of diabetes was given if the subject had both FPG ≥ 126 mg/dL and HbA1c ≥ 6.5% outside the reference range. Individual indexes to predict fatty liver were calculated as follows:

ZJU index = BMI (kg/m^2^) + FPG (mmol/L) + TG (mmol/L) + 3 × ALT (IU/L)/AST (IU/L) ratio (+2, if female) [21].

HSI = 8 × ALT/AST ratio + BMI (+2, if DM; +2, if female) [22].

FLI = (e^0.953*loge(TG) + 0.139*BMI + 0.718*loge(GGT) + 0.053*WC − 15.745^)/(1 + e^0.953*loge(TG) + 0.139*BMI + 0.718*loge(GGT) + 0.053*WC − 15.745^) * 100 [23].

The fibrosis-4 (FIB-4) index is a useful non-invasive index for the evaluation of liver fibrosis of chronic liver disease including NAFLD and is considered to have high diagnostic ability [29,30,31]. The FIB-4 index is calculated as [age (yr) × AST (U/L))/(Platelet count (10^9^/L) × √ALT (U/L)] [29]. The risk of advanced liver fibrosis (stage 3 or 4 according to Kleiner’s classification [32]) was evaluated using the FIB-4 index: low risk, FIB-4 index <1.3; intermediate risk, FIB-4 index 1.3–2.67, and high risk, FIB-4 index > 2.67 [30,31].

### 2.3. Abdominal Ultrasound Protocol and Definition of Fatty Liver

All subjects underwent abdominal ultrasonography to evaluate for fatty liver. The examination of all visible liver parenchyma was performed with a conventional convex array transducer. Liver parenchyma was examined with sagittal as well as longitudinal guidance of a probe and completed by lateral and intercostals views. The use of tissue harmonic imaging with both transducers was encouraged. The presence of steatosis was recognized as a marked increase in hepatic echogenicity, poor penetration of the posterior segment of the right lobe of the liver, and poor or no visualization of the hepatic vessels and diaphragm. The liver was considered normal if the hepatic parenchyma was homogeneous with no acoustic attenuation, the portal veins were visible, the diaphragm was well visualized, and echogenicity was similar to or slightly higher than that of the renal cortex. The study was performed using a LOGIQ 7 diagnostic imaging system with a 4 MHz convex array transducer (GE Healthcare, Waukesha, WI, USA), at Eguchi Hospital; a ProSound Alpha 10 diagnostic ultrasound system with a 3.5 MHz convex array transducer (Hitachi Aloka Medical, Ltd., Tokyo, Japan) at Kawamura Clinic Health Center; and a Xario ultrasound system, with a 3.5 MHz convex array transducer (Toshiba Medical Systems, Tochigi, Japan), at Kochi Medical School. The examinations were performed by sonographers with at least 5 years of experience, and who were trained by gastroenterologists with more than 5 years of experience. The technical parameters were adjusted for each subject using standard ultrasonography protocols, as previously reported [17,27]. Each certified gastroenterologist independently reviewed the images and evaluated the liver for the presence of steatosis. A semi-quantitative index (e.g., Hamaguchi et al. [33]) was not used for the grading of the severity of steatosis with careful consideration of the error due to the different ultrasonography equipment and examiners.

### 2.4. Statistical Analysis

Differences between the two groups were compared using the Mann–Whitney U-test. The predictive power of the indexes for detecting NAFLD was evaluated using the area under the receiver operating characteristic curves (AUROCs) with 95% confidence intervals (CIs). Comparisons of the AUROC among the indexes were performed using the DeLong test [34]. Sensitivities, specificities, positive predictive values, and negative predictive values were also calculated using the low cut-off value and high cut-off value: 30 and 36 for the HSI [22], 32 and 38 for the ZJU index [21] and 30 and 60 for the FLI [23]. A logistic regression model was used for the multivariate analysis, and all statistical differences were considered significant at *p* < 0.05. All analyses were performed using JMP Pro 14 (SAS Institute Inc., Cary, NC, USA).

## 3. Results

### 3.1. Characteristics of the Subjects

The characteristics of the subjects are summarized in Table 1. The study population consisted of 3316 males (47.8%) and 3611 females (52.2%) with a median age of 50.0 years. The median BMI and WC were 22.3 kg/m^2^ and 81.4 cm, respectively. Fatty liver was diagnosed in 1935 (28%) subjects.

### 3.2. Frequency Distribution of Individual Indexes

The frequency distribution of the HSI, ZJU index, and FLI is shown in Figure 1. All the indexes showed nonnormal distribution. Highly probable NAFLD was identified in 18%, 14%, and 9%, respectively, of the subjects using the HSI, ZJU index, and FLI individually. However, 43% of the subjects were not considered to have NAFLD according to the HSI and ZJU index, and 71% of the subjects were not considered to have NAFLD according to the FLI.

### 3.3. Comparison of HSI, ZJU Index, and FLI

The diagnostic performance of the indexes is compared in Figure 2. The AUROC was 0.874 (95% CI: 0.865–0.883) for the HSI, 0.886 (95% CI: 0.877–0.894) for the ZJU index, and 0.884 (95% CI: 0.876–0.892) for the FLI. The AUROC of the ZJU index and FLI were significantly greater than the HSI (vs. the ZJU index, *p* < 0.0001; vs. the FLI, *p* = 0.002). There was no significant difference between the ZJU index and FLI (*p* = 0.632). The sensitivities, specificities, positive predictive values, and negative predictive values of the indexes are summarized in Table 2. Using the high cut-off value (>36), the HSI detected NAFLD with 94.4% specificity and a 77.6% positive predictive value. The HSI excluded NAFLD with 93.4% sensitivity and a 95.7% negative predictive value using the low cut-off value (<30). The ZJU index detected NAFLD with 96.4% specificity and an 81.7% positive predictive value using the high cut-off value (>38). The ZJU index excluded NAFLD with 94.2% sensitivity and a 96.2% negative predictive value using the low cut-off value (<32). The FLI detected NAFLD with 98.4% specificity and an 87.5% positive predictive value using the high cut-off value (>60). The FLI excluded NAFLD with 68.8% sensitivity and an 87.7% negative predictive value using the low cut-off value (<30). Taken together, sensitivity with the low cut-off value and specificity with the high cut-off value were all higher than 90%, except sensitivity obtained with the low cut-off value of the FLI (68.8%).

### 3.4. Diagnostic Performance in Patients with Potential Advanced Liver Fibrosis

The subjects were stratified according to the advanced liver fibrosis risk evaluated with the FIB-4 index and ROC curve of the individual indexes (Figure 3). In the subjects with a low risk of advanced fibrosis, the AUROC was 0.888 (95% CI: 0.878–0.897) for the ZJU index and 0.892 (95% CI: 0.882–0.901) for the FLI, which was significantly greater than the HSI (0.881, 95% CI: 0.871–0.891; vs. the ZJU index, *p* = 0.002; and vs. the FLI, *p* = 0.007). In the subjects with an intermediate risk of advanced fibrosis, the AUROC was 0.860 (95% CI: 0.840–0.878) for the HSI, 0.870 (95% CI: 0.850–0.887) for the ZJU index, and 0.865 (95% CI: 0.846–0.882) for the FLI. The AUROC of the ZJU index was significantly greater than the HSI (*p* = 0.018). In the subjects with a high risk of advanced fibrosis, the AUROC was 0.888 (95% CI: 0.746–0.955) for the HSI, 0.912 (95% CI: 0.791–0.966) for the ZJU index, and 0.928 (95% CI: 0.816–0.974) for the FLI. There were no significant differences between the indexes. The sensitivity, specificity, positive predictive value, and negative predictive value of individual indexes are summarized in Table 3. The tendency of the diagnostic performance was similar with the analysis in the overall subjects; sensitivity with the low cut-off value and specificity with the high cut-off value were all around 90% regardless of the risk of advanced liver fibrosis, except the sensitivity with the low cut-off value of the FLI, which was lower than the ZJU index and the HSI in any categories of advanced fibrosis risk. However, the specificity of the FLI was the highest in any categories of advanced fibrosis risk.

### 3.5. Comparison of the ZJU Index and the FLI

Due to the AUROC of the ZJU index and the FLI being greater than the HSI in the overall subjects, we compared these two indexes, stratifying the subjects by gender, diabetes diagnosis, and ALT level (Figure 4). When the subjects were stratified by gender, the ZJU index showed a greater AUROC than the FLI in both female and male subjects. In females, the AUROC was 0.905 (95% CI: 0.893–0.917) for the ZJU index and 0.895 (95% CI: 0.881–0.907) for the FLI (*p* = 0.005). In males, the AUROC was 0.865 (95% CI: 0.853–0.877) for the ZJU index and 0.846 (95% CI: 0.832–0.859) for the FLI (*p* < 0.001). However, the diagnostic performance of the ZJU index was attenuated in the patients with diabetes. In the subjects without diabetes, there was no significant difference between the indexes: the AUROC was 0.881 (95% CI: 0.872–0.890) for the ZJU index and 0.883 (95% CI: 0.874–0.891) for the FLI (*p* = 0.574). In the subjects with diabetes, the AUROC of the FLI (0.861, 95% CI: 0.804–0.904) was significantly greater than that of the ZJU index (0.804, 95% CI: 0.739–0.856, *p* = 0.01). With regard to the subjects who were both within the reference range and outside the reference range of ALT, there was no significant difference in the AUROC between the ZJU index and the FLI. In the subjects with ALT ≤ 30 U/L, the AUROC was 0.859 (95% CI: 0.847–0.869) for the ZJU index and 0.862 (95% CI: 0.851–0.872) for the FLI (*p* = 0.463). In the subjects with ALT > 30, the AUROC was 0.856 (95% CI: 0.830–0.879) for the ZJU index and 0.842 (95% CI: 0.815–0.866) for the FLI (*p* = 0.157).

### 3.6. Characteristics of the Subjects with NAFLD Subjects Having a Negative ZJU Index and FLI

In the 2922 subjects showing both a negative ZJU index (<32) and FLI (<30), there were 107 subjects with NAFLD, and their characteristics were compared with 2815 subjects without NAFLD (Table 4). There were significant differences between the two groups in gender, age, BMI, WC, blood pressure, AST, ALT, GGT, FPG, TG, HDL-C, and LDL-C. According to the multivariate analysis, gender (male), BMI (>22 kg/m^2^), and abdominal circumference (male > 85 cm, female > 90 cm) were independently associated with NAFLD (Table 5).

## 4. Discussion

Noninvasive prediction formulas (HSI, ZJU index, and FLI) were tested in the current study in the identification of NAFLD. Using these indexes, NAFLD could be diagnosed accurately without an imaging examination in Japanese subjects who received health checkups. Ultrasound is the gold standard and most common imaging examination to diagnose fatty liver; however, with the high prevalence of NAFLD [8], it is impossible to recommend ultrasound for all in a general population. These indexes enable the identification of people with NAFLD in a large population who should receive an imaging examination.

The available guidelines, however, have never confirmed the actual screening procedure to identify NAFLD in the high-risk population, including diabetes patients, much less in the general population. This is because of uncertainties in diagnostic tests and treatment options, along with a lack of evidence related to the long-term benefits and cost-effectiveness of screening [35,36]. The guideline by the European Association for the Study of the Liver, European Association for the Study of Diabetes, and European Association for the Study of Obesity, regarding the utility of nonimaging biomarkers, including the FLI, is referred to for the screening of a large population [36]. Byrne and Targher recommended the use of the FLI as well as ultrasound for the screening of NAFLD in patients with type 2 diabetes [37]. Taken together, in the global “pandemic” of NAFLD, easy and low-cost screening procedures such as prediction formulas are warranted and should be promoted. In addition, general and common parameters are preferred as components of prediction formulas. According to the availability of the parameters, the appropriate prediction formula and nonimaging indexes should be used. Indexes analyzed in the current study comprised only general and common parameters: BMI, FPG, TG, and ALT for the ZJU index; BMI, AST, ALT, and diabetes for the HSI; and TG, BMI, GGT, and WC for the FLI. In the primary care setting and health checkup sites, where these parameters could be measured, any of the indexes should be tested to identify potential NAFLD patients.

There were several differences among the HSI, ZJU index, and FLI in the current study. In comparing the ROC of individual indexes, the ZJU index and FLI showed a significantly greater AUROC than the HSI. A possible explanation for the difference is the diagnosis of diabetes required for the HSI. According to the original HSI study by Lee et al., the diagnosis of diabetes was based on the FPG, HbA1c, and antidiabetic medications [22], whereas medication information was missing in our study. However, the sensitivity and specificity obtained in our current study (at a cut-off value of 30.0, sensitivity and specificity were 93.4% and 56.8%, respectively; at a cut-off value of 36.0, sensitivity and specificity were 49.4% and 94.4%, respectively) were comparable with the original study (at a cut-off value of 30.0, the sensitivity and specificity were 92.5% and 40.0%, respectively; at a cut-off value of 36.0, sensitivity and specificity were 46.0% and 92.4%, respectively). These results suggest that the diagnostic performance of the HSI was validated in the Japanese general population of the current study—as were the ZJU index and FLI. The distribution of the FLI was quite different from the other indexes (Figure 1); the frequent range of the index (peak of the distribution curve) shifted to the negative direction and the number of the subjects with an intermediate probability of NAFLD and a high probability of NAFLD were fewer than in the HSI and ZJU index. This unique distribution of the FLI might have resulted in a higher specificity than the HSI and ZJU index in our current study: 98.4% in the overall subjects (Table 2). However, the sensitivity of the FLI was lower than that of other indexes. According to the previous studies, the cut-off values of the FLI were optimized in Asia. Yang et al. reported from Taiwan that the optimal cut-off value to rule in an NALFD diagnosis by ultrasound was 35 for males and 20 for females, and the cut-off value to rule out the diagnosis was 25 for males and 10 for females [38]. According to another report from Taiwan, for a sensitivity ≥ 90%, the cut-off value was 15 for males and 5 for females, and for a specificity ≥ 90%, the cut-off value was 50 for males and 25 for females [39]. These cut-off values were lower than in the original study reported by Bedogni et al. from Italy [23], suggesting that the cut-off value, especially the low cut-off value for higher sensitivity, should be optimized in the Asian cohort. Without optimization of the original cut-off values, our results suggest that the HSI and ZJU index represented ≥90% sensitivity in the Japanese cohort and would be better for screening in the general population in Asia.

Among the noninvasive tests to diagnose the liver fibrosis of NAFLD, the FIB-4 index is a common and easy-access procedure that is calculated using AST, ALT, platelet count, and age [29,31]. Moreover, liver fibrosis is the most important finding to predict prognosis and to identify the treatment indication [24,25,40]. Therefore, screening with the approach “FIB-4 index first,” ahead of the diagnosis of fatty liver could be a novel and upcoming strategy to identify greater-risk NALFD in the primary care setting and at health checkups [38,41]. Hence, we tested the diagnostic performance of the HSI, ZJU index, and FLI in the subjects stratified by the FIB-4 index (Figure 3 and Table 3). The diagnostic performances of the HSI, ZJU index, and FLI in the individual FIB-4 index categories were similar to those of the overall subjects. Moreover, the diagnostic performance of the FLI increased in the subjects with an intermediate or high risk of advanced fibrosis. Taken together, the combination of the FIB-4 index and any of the ZJU index, HSI, and FLI would be useful to simultaneously predict the NAFLD and fibrosis risk.

In the current study, the ZJU index and FLI, which showed a greater AUROC than the HSI in the subjects, overall were compared under several specific conditions (Figure 4). Either in males or females, the ZJU index showed a greater AUROC than the FLI. The ZJU index reflects a gender difference in the formula, but the FLI does not [21,23], whereby the ZJU index might show a greater AUROC than the FLI when the subjects are divided by gender. However, whereas the ZJU index reflects FPG, the AUROC in subjects with diabetes was smaller than for the FLI, which does not include FPG, HbA1c, and the diagnosis of diabetes. This feature suggests that the interaction between the specific condition and individual components of the formula could both positively and negatively affect the diagnostic performance of the formula. According to our results, at least the ZJU index could be recommended for male only/female only subjects, and the FLI could be suitable for subjects with diabetes.

Adequate diagnostic performance of the ZJU index and FLI raised the question about the characteristics of the subjects with NAFLD having both a negative ZJU index and FLI. These false negative subjects should be carefully managed to avoid missing imaging examinations. In the current study, 107 subjects had NAFLD with both a negative ZJU index and FLI (Table 4). The median of the BMI, WC, liver enzymes, and metabolic parameters were all within the reference range but significantly higher than in the subjects without NAFLD. By the multivariate analysis, male sex, BMI > 22 kg/m^2^, and abnormal WC were independent factors associated with NAFLD (Table 5). Interestingly, these variables are not the results of laboratory tests, suggesting that physical findings are important for male subjects with blood test values within the reference range, and for those who have abdominal obesity and/or a BMI higher than 22 kg/m^2^, imaging examination could be recommended.

There are several limitations in the current study. Since the subjects were health checkup examinees and they were 20–65 years of age, the diagnostic performance of the HSI, ZJU index, and FLI should be confirmed in adolescent or younger subjects, as well as in older subjects. Because NAFLD was diagnosed by ultrasound and the subjects never underwent liver biopsy or an imaging examination such as MRI, evaluation for fatty liver was not quantitative and inter-/intra-observer error could be present in the ultrasound diagnosis. According to the recent development of an ultrasound-based technique, the attenuation of ultrasound in the liver parenchyma can be measured and the severity of fatty liver is quantitatively represented; FibroScan (Echosens, Paris, France) equips a controlled attenuation parameter (CAP) [42,43], and attenuation coefficient (Hitachi, Tokyo, Japan) [44] and attenuation imaging (Canon Medical Systems Corporation, Otawara, Tochigi, Japan) [45] are installed on the B mode ultrasound. Using these ultrasound-based techniques for the diagnosis of fatty liver as the standard, the diagnostic performance of nonimaging indexes, including HSI, ZJU index and FLI, should be tested in further study. On the other hand, these ultrasound-based techniques are relatively new and need to be validated further. There is evidence that skin capsular distance, BMI, and several other factors affect the reliability of CAP [43]. The accessibility of these ultrasound-based techniques is limited in the general population and primary care settings. Therefore, nonimaging indexes should be developed and individual features of nonimaging indexes should be well known.

In conclusion, the HSI, ZJU index, and FLI are useful to diagnose NAFLD in Japanese health checkup examinees. According to the availability of the parameters and characteristics of the cohort, the appropriate index should be used for screening for NAFLD.

## Figures and Tables

**Figure 1 diagnostics-11-00132-f001:**
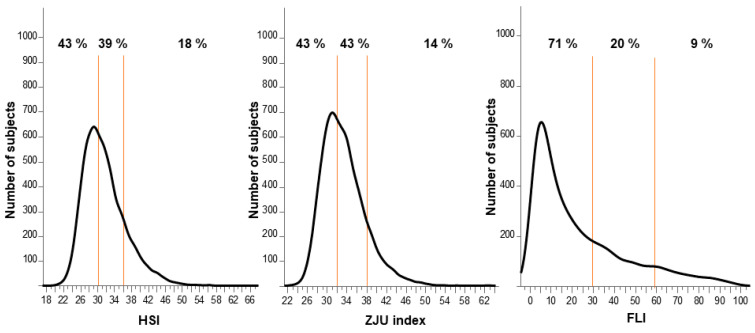
Frequency distribution of the HSI, ZJU index, and FLI. Graphs represent the frequency distribution of the HSI, ZJU index, and FLI. Orange lines represent cut-off values: 30 and 36 for the HSI, 32 and 38 for the ZJU index, and 30 and 60 for the FLI. The percentages in the graphs represent the proportion of subjects in each range. FLI, fatty liver index; HSI, hepatic steatosis index; ZJU, Zhejiang University.

**Figure 2 diagnostics-11-00132-f002:**
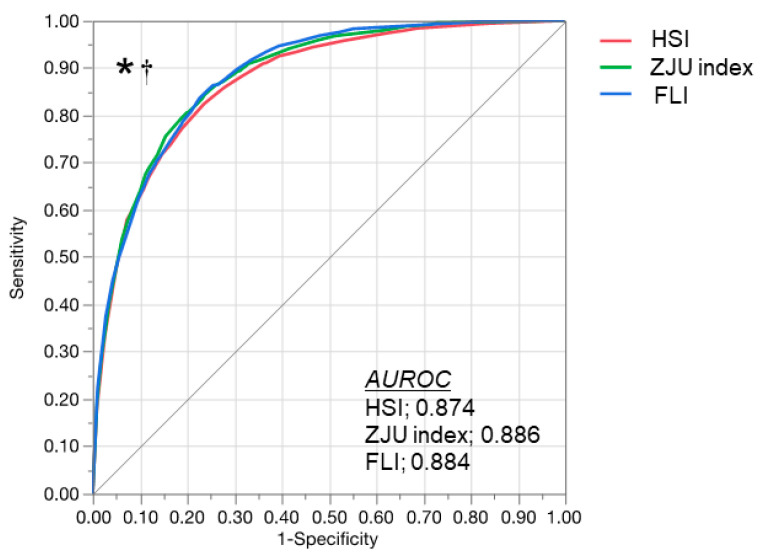
ROC curve of the HSI, ZJU index, and FLI for detecting NAFLD. ROC curve of the HSI (red), ZJU index (green) and FLI (blue) for the diagnosis of NAFLD in overall subjects. The ZJU index and FLI showed a greater area under the ROC curve than the HSI. * *p* < 0.05 in the comparison between the ZJU index and HSI. ^†^
*p* < 0.05 in the comparison between the FLI and HSI by the DeLong test. AUROC, area under the receiver operating characteristic; FLI, fatty liver index; HSI, hepatic steatosis index; NAFLD, nonalcoholic fatty liver disease; ROC, receiver operating characteristic; ZJU, Zhejiang University.

**Figure 3 diagnostics-11-00132-f003:**
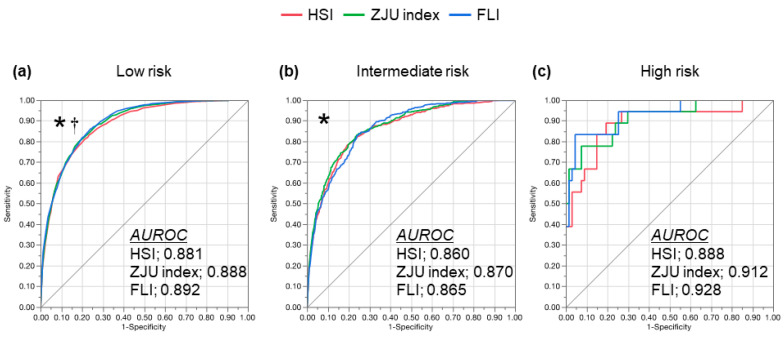
ROC curve of the HSI, ZJU index, and FLI for detecting NAFLD. ROC curve of the HSI (red), ZJU index (green) and FLI (blue) for the diagnosis of NAFLD stratified by the risk of advanced liver fibrosis (stage 3 or severe) evaluated using the FIB-4 index. (**a**) Low risk (FIB-4 index < 1.3); (**b**) intermediate risk (FIB-4 index 1.3–2.67); (**c**) high risk (FIB-4 index > 2.67). The ZJU index and FLI showed a greater AUROC than the HSI in low risk and the ZJU index showed a greater AUROC than the HSI in intermediate risk. * *p* < 0.05 in comparison between the ZJU index and HSI. ^†^
*p* < 0.05 in the comparison between FLI and HSI by the DeLong test. AUROC, area under the receiver operating characteristic; FIB-4, fibrosis 4; FLI, fatty liver index; HSI, hepatic steatosis index; NAFLD, nonalcoholic fatty liver disease; ROC, receiver operating characteristic; ZJU, Zhejiang University.

**Figure 4 diagnostics-11-00132-f004:**
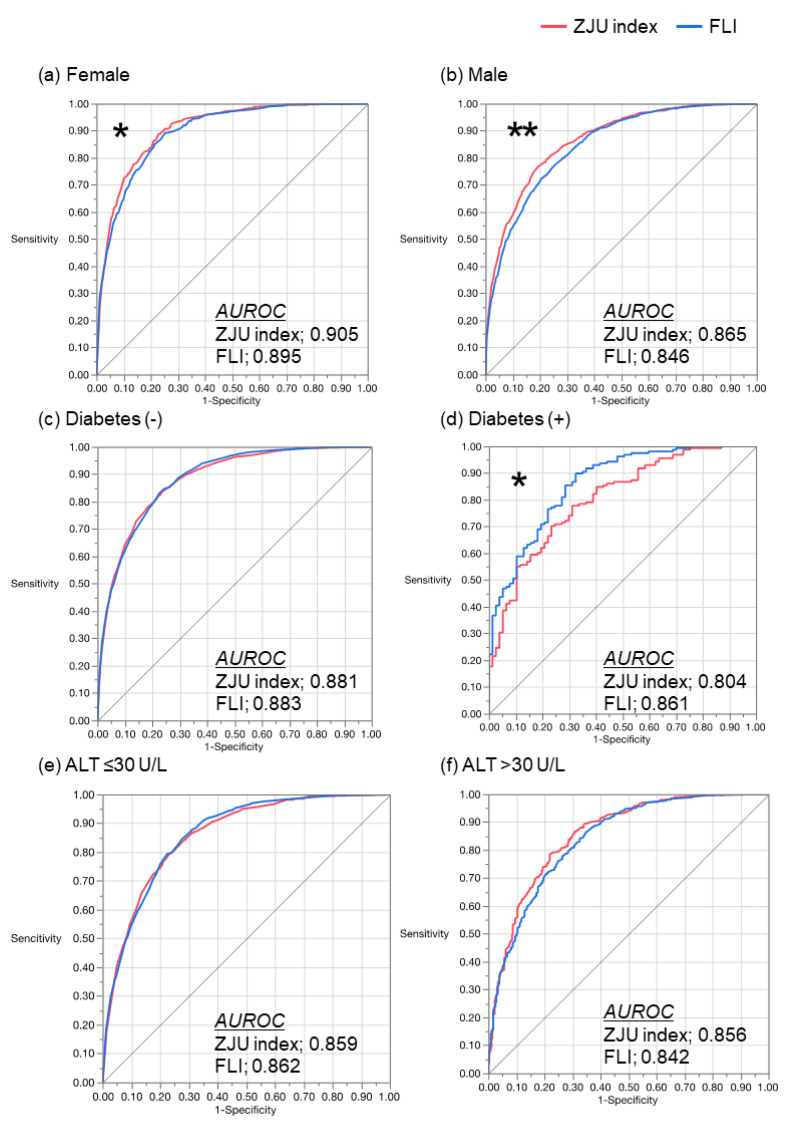
Comparison between the ZJU index and FLI under specific conditions. ROC curve of the ZJU index (red) and FLI (blue) for diagnosis of NAFLD. The ZJU index and FLI were compared in (**a**) females, (**b**) males, (**c**) subjects without diabetes, (**d**) subjects with diabetes, (**e**) subjects with ALT within the reference range (≤30 U/L), and (**f**) subjects with ALT > 30 U/L. The ZJU index showed a greater AUROC than the FLI when the subjects were stratified by gender. The FLI showed a greater AUROC than the ZJU index in the patients with diabetes. * *p* < 0.05. ** *p* < 0.001 by the DeLong test. AUROC, area under the receiver operating characteristic; FLI, fatty liver index; NAFLD, nonalcoholic fatty liver disease; ZJU, Zhejiang University.

**Table 1 diagnostics-11-00132-t001:** Characteristics of subjects.

	Total (*n* = 6927)
Age, years	50.0 (42.0–56.5)
Male, *n* (%)	3316 (47.8)
BMI, kg/m^2^	22.3 (20.2–24.4)
Waist circumference, cm	81.4 (75.0–87.0)
Platelet counts, ×10^4^/μL)	21.3 (18.4–24.5)
AST, U/L	19.0 (16.0–23.0)
ALT, U/L	18.0 (13.0–25.0)
ALP, U/L	202 (167–247)
GGT, U/L	23.0 (16.0–36.0)
FPG, mg/dL	96.0 (91.0–104)
TC, mg/dL	204 (182–228)
TG, mg/dL	88.0 (63.0–129)
HDL-C, mg/dL	60.0 (50.0–73.0)
LDL-C, mg/dL	120 (100–140)
HbA1c, %	5.55 (5.35–5.86)
Fatty liver, *n* (%)	1935 (28.0)

Continuous values are shown as median (lower and upper quartile). ALT, alanine aminotransferase; AST, aspartate aminotransferase; BMI, body mass index; FPG, fasting plasma glucose; GGT, γ-glutamyl transpeptidase; HbA1c, hemoglobin A1c; HDL-C, high-density lipoprotein cholesterol; LDL-C, low-density lipoprotein cholesterol; TC, total cholesterol; TG, triglyceride.

**Table 2 diagnostics-11-00132-t002:** Diagnostic accuracy of the HSI, ZJU index, and FLI.

Index	Cut-Off Point	Sensitivity (%)	Specificity (%)	Sensitivity + Specificity (%)	PPV (%)	NPV (%)
HSI	>36	49.4	94.4	143.8	77.6	82.8
	<30	93.4	56.8	150.2	45.6	95.7
ZJU index	>38	40.6	96.4	137.0	81.7	80.6
	<32	94.2	57.5	151.7	46.5	96.2
FLI	>60	28.2	98.4	126.6	87.5	77.8
	<30	68.8	87.0	155.8	67.6	87.7

Diagnostic performance of individual indexes using high cut-off value and low cut-off value. FLI, fatty liver index; HSI, hepatic steatosis index; NPV, negative predictive value; PPV, positive predictive value; ZJU, Zhejiang University.

**Table 3 diagnostics-11-00132-t003:** Diagnostic accuracy of the HSI, ZJU index, and FLI by the FIB-4 index.

Advanced Fibrosis	Index	Cut-Off Point	Sensitivity (%)	Specificity (%)	Sensitivity + Specificity (%)	PPV (%)	NPV (%)
Low risk	HSI	>36	54.4	93.4	147.8	77.8	83.0
		<30	95.3	52.8	148.1	45.9	96.4
	ZJU index	>38	42.6	95.9	138.5	81.8	79.7
		<32	95.8	56.1	151.9	48.3	96.9
	FLI	>60	29.5	98.4	127.9	89.1	76.6
		<30	70.4	86.8	157.2	69.6	87.3
Intermediate risk	HSI	>36	34.9	96.7	131.6	78.1	81.7
		<30	93.4	56.8	150.2	45.6	95.7
	ZJU index	>38	34.7	97.4	132.1	81.8	81.7
		<32	89.1	60.7	149.8	43.0	94.4
	FLI	>60	24.3	98.2	122.5	82.0	79.6
		<30	63.9	86.8	150.7	61.7	87.9
High risk	HSI	>36	38.8	98.5	137.3	87.5	85.7
		<30	88.8	77.6	166.4	51.6	96.3
	ZJU index	>38	44.4	100	144.4	100	87.0
		<32	94.4	61.1	155.5	39.5	97.6
	FLI	>60	38.8	100	138.8	100	85.9
		<30	83.3	94.0	177.3	78.9	95.5

Risk of advanced liver fibrosis was evaluated according to the FIB-4 index; low risk (FIB-4 index <1.3), intermediate risk (FIB-4 index 1.3–2.67), and high risk (FIB-4 index >2.67). FIB-4, fibrosis-4; FLI, fatty liver index; HSI, hepatic steatosis index; NPV, negative predictive value; PPV, positive predictive value; ZJU, Zhejiang University.

**Table 4 diagnostics-11-00132-t004:** Comparison of the characteristics between subjects with NAFLD and without NAFLD with a negative ZJU index and FLI.

	NAFLD+ *n* = 107	NAFLD– *n* = 2815	*p*-Value
Age, years	51 (43–59)	47 (40–55)	<0.001
Male, *n* (%)	80 (74.7)	959 (34.0)	<0.001
BMI, kg/m^2^	21.3 (20.5–22.1)	19.9 (18.8–20.9)	<0.001
Waist circumference, cm	80.8 (78–84)	74.2 (70.2–78.4)	<0.001
Systolic blood pressure, mmHg	111 (102–121)	105 (96–115)	<0.001
Diastolic blood pressure, mmHg	65 (61–73)	63 (57–71)	0.021
AST, U/L	20 (17–23)	18 (16–22)	0.013
ALT, U/L	17 (13–21)	14 (11–18)	<0.001
ALP, U/L	194 (157–248.5)	189.5 (155–232)	0.224
GGT, U/L	22 (17–30)	17 (13–25)	<0.001
FPG, mg/dL	95 (90–100)	93 (88–98)	<0.001
TC, mg/dL	197 (181–217)	200 (179–224)	0.361
TG, mg/dL	92 (66–115)	67 (52–89)	<0.001
HDL-C, mg/dL	57 (50–64)	69 (59–79)	<0.001
LDL-C, mg/dL	123 (105–136)	112 (93–131)	0.003
HbA1c, %	5.55 (5.35–5.75)	5.45 (5.35–5.65)	0.390

In the 2922 subjects with ZJU index < 32 and FLI < 30, NAFLD was diagnosed in 107 subjects by ultrasound. ALT, alanine aminotransferase; AST, aspartate aminotransferase; FLI, fatty liver index; FPG, fasting plasma glucose; HbA1c, hemoglobin A1c; HDL-C, high-density lipoprotein cholesterol; HSI, hepatic steatosis index; LDL-C, low-density lipoprotein cholesterol; TC, total cholesterol; TG, triglyceride; ZJU, Zhejiang University.

**Table 5 diagnostics-11-00132-t005:** Multivariate analysis to detect the factors associated with NAFLD in the subjects with a negative ZJU index and FLI.

	Odds Ratio	95% CI	*p*-Value
Gender (male)	3.97	2.46–6.40	<0.001
BMI (>22 kg/m^2^)	2.16	1.31–3.57	0.002
Waist circumference (>85 cm in males and >90 cm in females)	4.10	2.13–7.86	<0.001
Systolic blood pressure (≥130 mmHg)	1.22	0.56–2.64	0.612
Diastolic blood pressure (≥80 mmHg)	0.74	0.33–1.62	0.454
ALT (>30 U/L)	1.29	0.50–3.34	0.594
FPG (≥110 mg/dL)	1.36	0.56–3.25	0.488
TG (≥150 mg/dL)	1.62	0.66–3.93	0.285
HDL-C (<40 mg/dL)	1.54	0.43–5.50	0.501
LDL-C (≥140 mg/dL)	1.12	0.67–1.84	0.655

In the 2912 subjects with ZJU index < 32 and FLI < 30, NAFLD was diagnosed in 107 subjects by ultrasound and the association between the characteristics and NAFLD diagnosis was tested by the logistic regression model. BMI, body mass index; CI, confidence interval; ALT, alanine aminotransferase; FLI, fatty liver index; FPG, fasting plasma glucose; HDL-C, high-density lipoprotein cholesterol; HSI, hepatic steatosis index; LDL-C, low-density lipoprotein cholesterol; NAFLD, nonalcoholic fatty liver disease; TG, triglyceride; ZJU, Zhejiang University.

## Data Availability

The data presented in this study are available on request from the corresponding author. The data are not publicly available due to the Japanese Clinical Trials Act (https://www.mhlw.go.jp/stf/seisakunitsuite/bunya/hokabunya/kenkyujigyou/i-kenkyu/index.html).

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
