# Peer review of "Prediction of Nonalcoholic Fatty Liver Disease Using Noninvasive and Non-Imaging Procedures in Japanese Health Checkup Examinees"

_diagnostics, 2021, doi:10.3390/diagnostics11010132_

Round 1
Reviewer 1 Report
The manuscript is well written, but unfortunately it has nothing new (even for Japan) to bring to the literature.
Therefore, it should be rejected.
Author Response
Point 1: The manuscript is well written, but unfortunately it has nothing new
(even for Japan) to bring to the literature. Therefore, it should be rejected.
Response 1: In the present study, we compared the 3 non-imaging indexes (ZJU index, HSI and FLI) to predict fatty liver. To date, there has been no study comparing the diagnostic performance of these 3 indexes simultaneously in the relatively large number of the sunjects who performed ultrasound. Moreover, we tested diagnostic performance of ZJU and FLI in the stratified subjects by the FIB-4 index. As we described in the manuscript, screening to rule out/rule in the advanced fibrosis is the hot topic because it is well recognized that liver fibrosis is the most significant pathological finding which associated with the prognosis of NAFLD. FIB-4 index is one of the common and well validated non-invasive procedures to evaluate liver fibrosis. Therefore, numerous epidemiological studies about FIB-4 index have been reported and several guidelines of NAFLD/NASH recommend to use FIB-4 index. However, generally, presence of fatty liver should be diagnosed ahead of the evaluation of the liver fibrosis in the prehospital population like examinee of health check-up. It must be a serious concern if the diagnostic performance of non-imaging indexes would decrease in the patients who have advanced liver fibrosis risk like FIB-4 index > 2.67. Present study, at least partly, dispelled this concern and to our knowledge, there has been no similar study about non-imaging indexes which included the association with FIB-4 index. Finally, we performed literature search again to find a Japanese epidemiological study about FLI and/or ZJU index and/or HSI which referred to ultrasound as the gold standard. One multicentre epidemiological survey used the FLI and included the Japanese cohort to estimate the prevalence of fatty liver, but it was not the study to validate the diagnostic performance of FLI in Japan (Jensen T, et al. Sci Rep. 2018 6;8:11735). Other Japanese studies used FLI as a predictor of the incidence of diabetes (Nishi T et al. J Diabetes Investig. 2015;6:309-16. Hirata A et al. Hepatol Res. 2018;48:708-716). There are 3 studies which used FLI to evaluate the effect of sodium glucose cotransporter 2 inhibitor (ipragliflozin) on fatty liver, but these are not the studies to test the diagnostic performance of FLI in Japan. Also, there has not been the studies which tested the diagnostic performance of ZJU index and HIS in Japan. In the guideline, “EASL-EASD-EASO Clinical Practice Guidelines for the management of non-alcoholic fatty liver disease” published in 2016 summarized the information about FLI and HSI in Supplementary Table 3. According to the guideline, no validation study of FLI and HSI is available in Japan at least up to April 2015. We performed literature search again using PubMed and confirmed that there has been no validation study in Japan. It is grateful if the reviewer would suggest the literature from Japan.
Taken together, we think that present study includes novel points below.
- Comparison of multiple non-imaging indexes
- Test the association with liver fibrosis evaluated by the FIB-4 index
- Validation in the Japanese cohort
Thank you so much for the comment.
Reviewer 2 Report
The study is interesting, however there are some concerns in terms of reliability of non-invasive scores to predict steatosis and the employement of ultrasound without the gold standard of liver biopsy (or at least elastography).
Also, why a ultrasonographic steatosis score was not used? (e.g. Hamaguchi score)
For this reason, I suggest citing in the introduction this paper that reports the use of non-invasive scores if compared to elastography: DOI: https://doi.org/10.1016/j.aohep.2020.04.003
Also, the authors should specify (to strength their thesis) that elastography in NAFLD patients is not always reliable, please cite: https://doi.org/10.3390/diagnostics10100795 (patients with higher skin-to-liver distance have low reproducibility), and the impact of serum transaminases: 1- https://doi.org/10.3390/microorganisms8030348 2 - 10.1007/s40477-020-00456-9
Author Response
Response to Reviewer 2 Comments
Thank you so much for the positive comments. We revised the manuscript accordingly and believe that our manuscript has improved.
Point 1: The study is interesting, however there are some concerns in terms of
reliability of non-invasive scores to predict steatosis and the employment of ultrasound without the gold standard of liver biopsy (or at least elastography).
Response 1: As reviewer suggested, liver biopsy was not performed in the present study because of the relatively large sample size of general population cohort was investigated. Recently, ultrasound-based technique enables to diagnose the fatty liver as well as MRI; these procedures are non-invasive, quantitative and reliable (43-46). Accordingly, we described this point in the limitation (page 11, line 364-371).
.
Point2: Also, why a ultrasonographic steatosis score was not used? (e.g.Hamaguchi score) For this reason, I suggest citing in the introduction this paper that reports the use of non-invasive scores if compared to elastography:
DOI: https://doi.org/10.1016/j.aohep.2020.04.003
Response 2: Thank you so much for the comment from the reviewer. Because there were multiple examiners for ultrasound in the individual institutions, we avoided to evaluate the severity of fatty liver according to the B mode imaging in terms of reliability and reproducibility. However, as reviewer suggested, ultrasonographic steatosis score should be cited in this manuscript (33). We modified introduction accordingly (page 3, line 129-131). Because our study was not about fibrosis, but about steatosis, we did not cite the paper suggested by the reviewer (DOI: https://doi.org/10.1016/j.aohep.2020.04.003).
Point 3: Also, the authors should specify (to strength their thesis) that elastography in NAFLD patients is not always reliable, please cite: https://doi.org/10.3390/diagnostics10100795 (patients with higher skin-to-liver distance have low reproducibility), and the impact of
serum transaminases: 1- https://doi.org/10.3390/microorganisms8030348
2 - 10.1007/s40477-020-00456-9
Response 3: Because our study was not about fibrosis, but about steatosis, it is difficult to cite the papers suggested the reviewer. However, it is helpful to consider the possible limitation of elastography (CAP) and affecting factors for CAP measurement which might decrease its reliability (47). We described this point in page 11, line 371-376.

Round 2
Reviewer 1 Report
After the author's response I suggest publication of the manuscript